# Video-Based Communication and Its Association with Loneliness, Mental Health and Quality of Life among Older People during the COVID-19 Outbreak

**DOI:** 10.3390/ijerph18126284

**Published:** 2021-06-10

**Authors:** Tore Bonsaksen, Hilde Thygesen, Janni Leung, Mary Ruffolo, Mariyana Schoultz, Daicia Price, Amy Østertun Geirdal

**Affiliations:** 1Department of Health and Nursing Science, Faculty of Social and Health Sciences, Inland Norway University of Applied Sciences, 2418 Elverum, Norway; 2Faculty of Health Studies, VID Specialized University, 4306 Sandnes, Norway; hilde.thygesen@oslomet.no; 3Prosthetics and Orthotics, Department of Occupational Therapy, Faculty of Health Sciences, Oslo Metropolitan University, 0130 Oslo, Norway; 4Faculty of Health and Behavioural Science, The University of Queensland, Brisbane, QLD 4072, Australia; j.leung1@uq.edu.au; 5School of Social Work, University of Michigan, Ann Arbor, MI 48109, USA; mruffolo@umich.edu (M.R.); daiciars@umich.edu (D.P.); 6Faculty of Health and Life Sciences, Northumbria University, Newcastle upon Tyne NE1 8ST, UK; mariyana.schoultz@northumbria.ac.uk; 7Department of Social Work, Faculty of Social Sciences, Oslo Metropolitan University, 0130 Oslo, Norway; amyoge@oslomet.no

**Keywords:** coronavirus, cross-national study, pandemic, physical distancing, psychological outcomes, social distancing

## Abstract

The aim of the study was to examine the use of video-based communication and its association with loneliness, mental health and quality of life in older adults (60–69 years versus 70+ years) during the COVID-19 pandemic. A cross-sectional online survey was conducted in Norway, UK, USA and Australia during April/May 2020, and 836 participants in the relevant age groups were included in the analysis. Multiple regression analyses were conducted to examine associations between the use of video-based communication tools and loneliness, mental health and quality of life within age groups, while adjusting by sociodemographic variables. Video-based communication tools were found to be more often used among participants aged 60–69 years (60.1%), compared to participants aged 70 or above (51.8%, *p* < 0.05). Adjusting for all variables, the use of video-based communication was associated with less loneliness (*β* = −0.12, *p* < 0.01) and higher quality of life (*β* = 0.14, *p* < 0.01) among participants aged 60–69 years, while no associations were observed for participants in the oldest age group. The use of video-based communication tools was therefore associated with favorable psychological outcomes among participants in their sixties, but not among participants in the oldest age group. The study results support the notion that age may influence the association between the use of video-based communication tools and psychological outcomes amongst older people.

## 1. Introduction

Current video-based communication tools use an internet connection to allow users to communicate in real-time and see each other while doing so. In recent years, several such tools have also been developed to enable digital sharing of materials between the persons taking part in the call. Thus, video-based communication tools can greatly enhance the possibilities for exchange as well as the general communication experience, in comparison to regular telephone calls. While the first video-based communication system was launched by AT&T (Dallas, USA) in 1964 [1], recent years have seen great improvements in the employed technologies, and there is much competition to attract users.

With the onset of the global COVID-19 pandemic in March 2020, many countries experienced a lockdown. People were generally instructed to practice ‘social distancing’ [2], which implied maintaining a physical distance to people outside the household and, as far as possible, to stay at home to prevent viral spread. Schools and nurseries were closed, as were many shops and businesses [3]. Flights and travels were cancelled, as were all sporting, religious and cultural events. Practically overnight, working from home and attending online classes became the new standard for many workers and students. In this situation, the worldwide use of video-based communication tools peaked exceptionally: in March 2020, Skype (Luxembourg City, Luxembourg) was reported to have 100 million monthly users and 40 million users daily, representing a 70% increase since the preceding month [4]. In April 2020, Zoom (San Jose, CA, USA) experienced 300 million daily meeting participants [5]. It seems fair to assume that a substantial part of the increase in the use of video-based communication tools was related to the communication needs of companies and employees working remotely. However, as the pandemic imposed radical restrictions on face-to-face social contact, video-based communication also became a viable way of maintaining personal contact in a situation where regular contact was difficult or even prohibited [6,7,8,9].

Before the pandemic outbreak, substantial research had addressed elevated levels of loneliness and its relationship to mental health and quality of life in older people [10,11,12]. For example, one study found that loneliness impacted on quality of life via two routes, mediated by mental health and mediated by resilience factors [13]. The study suggested that increasing social support, and also increasing resilience and reducing the emotional burden of loneliness, is important for health and quality of life in older age [13]. Thus, in the COVID-19 era, promoting mental health and preventing loneliness through the use of video-based technologies seems particularly relevant for older people who are restricting real-life social interactions because of a higher risk of severe illness or fatal outcome if exposed to the virus. However, studies have shown that people of older age may have barriers in access to information and communication technologies (ICTs) in general [14], and they may be less inclined to use them, regardless of purpose [14,15,16]. Thus, paradoxically, it appears that those who may experience the most personal benefit from using such communication technologies may be the least likely to adopt them.

Notwithstanding the potential of such technologies to reduce social and health inequities [9,17], and possibly to reduce isolation and increase quality of life among older people during the pandemic [18,19], previous studies on the use of video-based communication in the delivery of interventions targeting mental health have been ambiguous in their conclusions. While short-term success from remote therapy for mental health problems has been found, older people’s poorer access to and competence in using ICTs may hinder its implementation [20]. A review comprising three studies concerned with nursing home residents showed little to no evidence of improvements in loneliness, depression or quality of life due to the use of video-based communication [21], whereas other studies have concluded in favor of using new technologies (in a broader sense) to tackle social isolation and loneliness among older people [22,23,24]. Moreover, research on the association between use of ICTs in general and psychological outcomes have suggested that the very old may experience greater benefits, compared to those not so old. For example, Fang and co-workers [25] found that age moderated the association between use of ICTs and psychological well-being among the elderly. Only among those in the oldest age group (75+ years) was use of ICTs associated with higher well-being, and particularly so where ICTs facilitated contact with family members.

In view of the literature review above, those in the oldest age group may have a particularly restricted social life due to COVID-19. Thus, their use of video-based communication tools may have the potential to protect against loneliness and promote mental health and quality of life. However, poorer ICT competence and familiarity among the oldest may hinder the realization of this potential. Explorative research on the associations between the use of video-based communication and psychological outcomes in different age groups is therefore warranted. Increased knowledge in this area may inform healthcare workers, policymakers, and the general public about the role of video-based communication tools among older people.

The aim of this study was to examine the use of video-based communication and its association with loneliness, mental health, and quality of life in older adults aged 60–69 years versus 70+ years during the COVID-19 pandemic.

## 2. Materials and Methods

### 2.1. Procedure

An invitation to participate in this self-administered survey was distributed via different social media in Norway, USA, UK and Australia during April and May 2020 [26,27]. A narrow scope during data collection constitutes a risk of bias related to the demographic composition of the sample, and consequently, the sample may have limited ability to represent the general population [28]. Therefore, data were collected from all four countries where the researchers were based. Each country had a landing site for the survey at the researcher’s universities: OsloMet-Oslo Metropolitan University, Norway; University of Michigan, USA; University of Salford, UK; and the University of Queensland, Australia, respectively. The initiator of the project was AØG from OsloMet, but all countries and universities had their own head of the project, due to ethical considerations and permissions. The survey was translated from Norwegian to English by the researchers according to language and cultural contexts. To be included in the study, participants had to be 18 years or older, understand Norwegian or English and live in Norway, USA, UK, or Australia. Furthermore, to be included in the analyses of the current sub-study, participants were required to be 60 years or older.

### 2.2. Measures

#### 2.2.1. Sociodemographic Characteristics

Sociodemographic variables included age group (60–69 years versus 70 years and above), sex (male versus female), highest completed education level (high school, associated/technical degree or lower versus bachelor’s degree or higher), cohabitation (living with spouse or partner versus not), and employment status (having full-time or part-time employment versus not).

#### 2.2.2. Use of Video-Based Communication Platforms

The participants were asked to indicate (yes versus no) whether they used any of the following video-based communication platforms: FaceTime (Apple Inc.; Cupertino, CA, USA), Skype (Luxembourg City, Luxembourg), Zoom (San Jose, CA, USA), and Teams (Redmond, WA, USA). A categorical variable was created to distinguish between those who used at least one of these video-based communication platforms, and those who did not.

#### 2.2.3. Loneliness

The Loneliness Scale [29] consists of six statements, all of which are rated from 0 (totally disagree) to 4 (totally agree). It was designed to measure two different aspects of loneliness, social loneliness (e.g., “There are plenty of people I can rely on when I have problems”) and “emotional loneliness” (e.g., “I experience a general sense of emptiness”). Previous studies have suggested that the six statements may be used with two viable factor solutions: a two-factor solution reflecting social and emotional loneliness as two different aspects of loneliness or with a one-factor solution suggesting that all six items tap into one general construct of loneliness [29,30]. In this study, the one-factor solution was used. Cronbach’s α was 0.74 for the 6-item loneliness scale. The score range is 0–24, with higher scores indicating more loneliness.

#### 2.2.4. Mental Health

The General Health Questionnaire 12 (GHQ-12) is widely used as a self-report measure of mental health [31,32]. A large number of studies in the general adult population, clinical populations, work populations and student populations have provided support for its validity across samples and contexts [32,33,34,35]. Six items of the GHQ-12 are phrased positively (e.g., ‘able to enjoy day-to-day activities’), while six items are phrased as a negative experience (e.g., ‘felt constantly under strain’). On each item, the person indicates the degree to which the item content has been experienced during the two preceding weeks, using four response categories (‘less than usual’, ‘as usual’, ‘more than usual’ or ‘much more than usual’). Items are scored between 0 and 3, and positively formulated items are recoded prior to analysis. As a result, the GHQ-12 scale score range is 0–36, with higher scores indicating poorer mental health (more psychological distress). Cronbach’s α was 0.89 in this sample.

#### 2.2.5. Quality of Life

Cantril’s ladder (CL) is a self-administered overall quality of life (QoL) questionnaire with one question, “How is your life”, asking the person to rate his or her present experience of life on a scale anchored by their own identified values [36]. The response alternatives range between 0 and 10, with 0 = worst possible QoL and 10 = best possible QoL. Good QoL is often operationalized as having a CL score of six or above. The CL has been reported to have good validity and stability and reasonable reliability [37,38].

### 2.3. Statistical Analysis

Group proportions within sex, education level, cohabitation, employment status and use of video-based communications were compared by Chi-Square tests. Ratings on loneliness, mental health and quality of life were compared between participants in the two age groups by independent *t*-tests. Adjusted associations between independent variables and each of the outcome variables (loneliness, mental health and quality of life) were assessed with multiple linear regression analyses. Within each of the two age groups, ratings on the outcome variables were assessed in relationship to use or non-use of video-based communication platforms, while adjusting by sex, education level, cohabitation and employment status. Initial regression analyses also included the interaction term ‘video-based communication × country’ to assess whether associations between the use of video-based communications and the outcome variables differed by country. If the interaction term was not statistically significant, it was removed prior to the final analysis. In cases of statistically significant interactions, the analyses were repeated for each country separately. All independent variables were included in one step. Statistical significance was set at *p* < 0.05.

### 2.4. Ethics

The data in this cross-sectional and cross-country study were collected anonymously. All ethical rules were followed in each country. The study was thereby quality assured and approved by OsloMet (20/03676) and the regional committees for medical and health research ethics (REK; Ref. 132066) in Norway, reviewed by the University of Michigan Institutional Review Board for Health Sciences and Behavioral Sciences (IRB HSBS) and designated as exempt (HUM00180296) in the USA, by University Health Research Ethics (HSR1920-080) in the UK and by (HSR1920-0802020000956) in Australia.

## 3. Results

### 3.1. Participants

The characteristics of the sample are displayed in Table 1, with age groups comparisons. The sample consisted of 836 individuals in total: Norway (*n* = 93, 11.1%), USA (*n* = 417, 49.9%), UK (*n* = 273, 32.7%) and Australia (*n* = 53, 6.3%), and the larger proportion was aged 60–69 years (*n* = 612, 73.2%). The majority (75.7%) were women, and 70.2% had education at the bachelor’s degree level or higher. Among those aged 60–69 years (*n* = 612), full-time or part-time employment was held among 47.7%. Among those aged 70 years or more (*n* = 224), having employment was less common (16.5%, *p* < 0.001). The use of video-based communication was more common among those aged 60–69 (60.1%), compared to those in the oldest age group (51.8%, *p* < 0.05).

Table 1 also displays the levels of loneliness, mental health and QoL according to age groups. More mental health problems among those aged 60–69 years compared to those aged 70 or older (*M* = 15.1 *versus M* = 14.2, respectively, *p* = 0.05) bordered on statistical significance. For loneliness and QoL, the differences were not statistically significant.

### 3.2. Adjusted Associations between the Use of Video-Based Communication and Loneliness, Mental Health and Quality of Life

Adjustments were made for gender, education, employment and cohabitation status, and possible interactions between video-based communication and country were tested in initial models. Only one of the interaction terms was statistically significant, indicating that among those aged 70 or above, the association between video-based communication and quality of life differed between the four countries (*p* < 0.05). In all other analyses, the interaction term was not statistically significant and therefore removed. The results from the regression analyses are displayed in Table 2.

Among participants aged 60–69 years, use of video-based communication platforms was associated with lower ratings of loneliness (*β* = −0.12, *p* < 0.01) and higher QoL ratings (*β* = 0.14, *p* < 0.01). In the whole sample, among participants aged 70 years or older, use of video-based communication platforms was not significantly associated with any of the outcome measures. In the country-specific analyses, the associations between these variables were found to be both positive (in UK and Australia) and negative (in Norway and USA), but they were not statistically significant for any of the countries.

Across age groups, female gender was consistently associated with poorer mental health and QoL, while living with a spouse or partner was consistently associated with less loneliness and better QoL. Having employment was associated with better mental health among those aged 60–69 years. Each of the regression models accounted for small, but statistically significant proportions of the outcome variance (between 4.9% and 10.4%).

## 4. Discussion

This study examined the use of video-based communication and its association with loneliness, mental health and quality of life in older adults between two different age groups (60–69 years versus 70 years or above) during the COVID-19 pandemic. In summary, more than half of the participants in both groups used video communication, with the older group using video communication less often. Among those aged 70 or more, the use of video-based communications was not related to any of the psychological outcomes. Among those aged 60–69 years, in contrast, the use of video-based communication was significantly related to lower loneliness and higher quality of life, although with small effect sizes. Overall, while levels of loneliness and mental health problems were elevated, in comparison to pre-pandemic studies [39,40], there were similar levels of loneliness, mental health and quality of life across the two age groups.

Video-based communication was used by more than half of the participants in each age group but was used by a bigger proportion of those aged 60–69, compared to their older counterparts. The generally frequent use of video-based communication tools in both age groups may be interpreted as a reflection of the recruitment procedure. Recruitment by social media may have skewed the sample towards having more participants who are familiar with ICTs in general, including the use of video-based communication tools. However, the lower proportion of participants in the oldest age group using video-based communication tools may reflect the findings in prior studies [14,15,16], suggesting that the inclination to use video-based communication tools, and indeed ICTs in general, decreases in old age. This finding may also reflect a generational gap in digital competence, as those in the aged 70+ years are less likely to have been exposed to digital technologies in their previous employment and everyday life than younger generations.

In addition, a higher proportion of the 60–69 age group were still in employment compared to the 70 and older group, which may also account for the differences in use of video-based tools. With the onset of COVID-19, the use of video-based communication tools, such as Skype and Zoom, increased radically [4,5], and a substantial part of the increase was likely related to the communication needs of companies and employees working remotely.

Among those aged 70 years or older, the study showed no significant associations between use of video-based communications and any of the psychological outcome measures. In other words, their variations in loneliness, mental health and quality of life were not dependent on whether they used video-based communications. Thus, our findings support the view presented by Noone and co-workers [21], essentially stating no effect of videocall interventions to reduce loneliness in older adults. However, one should take into account the moderation effects found by Fang and co-workers [25], indicating that the association between ICT use in general and personal well-being was present only among those having contact with family members and only among frail participants. Thus, the role of the person with whom there is contact appears to matter a great deal. In short, if family members count and others do not, this qualification may explain poor effects of telehealth interventions implemented by volunteers or healthcare personnel. Moreover, the moderation effect of frailty status may indicate that video-based contact can be helpful, but only when the option of having face-to-face contact is reduced. Thus, if the oldest participants in our study were still able to have regular contact with other people, this may explain the lack of association between video-based communication and the psychological outcomes. In addition, retirement from work, which is commonly found in people over the age of 70 years, can also imply emotional disengagement with technologies that may have been strongly related to the worker role.

Among the oldest participants, it could also be that video-based contact needs to be initiated by family members such as children and grandchildren. In support of this reasoning, a qualitative study from Norway found that the sample of very old persons (median age 90.5 years) were predominantly “passive” users of video-based communication solutions (receiving calls, but not necessarily making calls themselves) [41]. This may be due to a lack of confidence and competence with the technological solutions, and fear of doing something wrong. Consequently, video-based communication may not be experienced as an available option when they feel lonely and in need of support.

Among those aged 60–69 years, using video-based communication tools was significantly associated with lower loneliness and better quality of life. Among participants in this group, almost 50% still had full-time or part-time employment. Therefore, the association might partly reflect that employed participants were able to use video-based tools to connect with their colleagues, students, collaborators and work partners in job-related activities, which in turn might protect against a sense of loneliness and reduced quality of life. For those in employment, regular video-based job meetings might contribute to cover their need for contact with people outside of their own household. Thus, their use of video-based communication tools may be logically related to better quality of life and less loneliness.

In comparison to the oldest participants, it could also be that participants who were in their sixties would be more comfortable with using ICTs in general [14,15,16] and were able to use them in a way that served their needs for connecting with the people who were important to them [25]. Moreover, feeling familiar with video-based communication tools may make the technologically mediated interactions feel more ‘real’, as opposed to the somewhat estranged feeling that may arise from using technologies that are unfamiliar.

The study was a sub-study of a larger cross-national comparative survey concerned with the impact of social distancing on mental health and quality of life during the early stages of the COVID-19 outbreak [26,27]. Recruitment to the study was performed by distributing the link to the web-based survey via social media; thus, participants in the study were likely to be regular social media users. Compared to the general population, the participants were largely female and highly educated. Therefore, the generalizability of the results to the general populations of the involved countries may be questioned. Our study may have missed older adults with lower education who may experience greater barriers with accessing video-based communication to keep connected during these challenging times.

Moreover, we did not collect data concerned with the participants’ competence and familiarity with the technologies used in video-based communication. Nor did we collect data on how such communication occurred, how often, for how long, with whom and with what purpose. Thus, several interesting aspects related to video-based communication among older people were not elucidated by our study.

Following the result of a significant interaction between video-based communication and quality of life among those aged 70 years or older, the sample size of participants in this age group may have been too small to detect significant associations when rerunning the analyses by country. There may be differences between countries because of different social distancing policies in place, with mental health consequences, to protect older adults as they are at a higher risk of severe consequences if they get infected with COVID-19. Future studies in this field of research may preferably ascertain sufficient group sizes to enable the investigation of whether associations vary between countries. They may also assess, more specifically, the details of video-based communication practice, i.e., how often, with whom and for what purposes they were used (e.g., work-related or personal use; contact with co-workers, business partners, family or friends).

The comparisons were between participants in two different age-groups. About 50% of the respondents in the 60–69 age group were still in work, and we wondered if this might have impacted on our results. A recent systematic review of general population studies found that employment protected against mental distress during the early stage of the pandemic [42]. While those in the oldest age group had marginally better mental health than those aged 60–69 years, we found that employment was associated with better mental health among those aged 60–69 years. However, the finding that the use of video-based communication was associated with lower loneliness and better quality of life was still valid (and a better predictor than employment) after adjusting for employment status.

## 5. Conclusions

This study examined the use of video-based communication and its association with loneliness, mental health and quality of life among older adults during the COVID-19 pandemic. A larger proportion of participants aged 60–69 years used video-based communication tools, compared to those aged 70 years or above. Using such tools was associated with less loneliness and higher quality of life among participants in their sixties, whereas no significant associations were found among participants in the oldest age group. The study results support the notion that age may influence the association between use of video-based communication tools and psychological outcomes amongst older people, whereas the nature of this influence was in contrast to findings in previous research. While differences in the use of video-based communication likely concern more work participation among participants aged 60–69 years, less familiarity with ICTs and possibly a stronger inclination to disregard their value among persons in the oldest age group may more likely explain the age-dependent associations between video-based communication and psychological outcomes.

## Figures and Tables

**Table 1 ijerph-18-06284-t001:** Sample characteristics with age group comparisons.

	Age 60–69 Years (*n* = 612, 73.2%)		Age 70 Years and above (*n* = 224, 26.8%)		
Characteristics	*n*	%	*n*	%	*p*
Sex					0.32
Male	140	23.0	58	26.2	
Female	470	77.0	163	73.8	
Education level					0.64
High school or lower	185	30.2	64	28.6	
Bachelor’s degree or higher	427	69.8	160	71.4	
Cohabitation					0.24
Yes	396	74.3	127	69.8	
No	137	25.7	55	30.2	
Employment					<0.001
Full-time or part-time	292	47.7	37	16.5	
No	320	52.3	187	83.5	
Video-based communication					
FaceTime	280	47.0	79	36.7	<0.05
Skype	197	33.1	48	22.1	<0.01
Zoom	87	14.9	27	12.7	0.42
Teams	33	5.7	8	3.8	0.27
At least one of the above	368	60.1	116	51.8	<0.05
Psychological factors	*M*	*SD*	*M*	*SD*	
Loneliness	9.3	4.4	9.1	4.4	0.58
Mental health	15.1	6.5	14.2	5.7	0.05
Quality of life	6.8	2.2	7.0	2.1	0.09

Note. Statistical tests are Chi-Square tests (categorical variables) and independent *t*-tests (continuous variables). Cohabitation refers to ‘living with a spouse or partner’.

**Table 2 ijerph-18-06284-t002:** Adjusted associations between the use of video-based communication and loneliness, mental health and quality of life within age groups.

	Age 60–69 (*n* = 612)			Age 70 and above (*n* = 224)		
Independent Variables	Loneliness	MH	QOL	Loneliness	MH	QOL
Female sex	0.02	0.14 **	−0.09 *	0.10	0.28 ***	−0.21 **
Having higher education	−0.01	−0.07	0.04	−0.13	0.06	−0.02
Spouse/partner	−0.21 ***	−0.07	0.16 ***	−0.16 *	−0.03	0.17 *
Having employment	−0.08	−0.14 ***	0.03	−0.06	−0.06	0.14
Use of video-based communication	−0.12 **	−0.07	0.14 **	−0.03	0.01	−0.04
Explained variance	6.6% ***	4.9% ***	6.1% ***	7.3% *	8.4% **	10.4% **

Note. MH is mental health, as measured with the General Health Questionnaire. QOL is quality of life, as measured with Cantril’s ladder. Table content is standardized *β* weights. Higher ratings on loneliness and QOL is more loneliness and higher QOL, while higher ratings on MH is poorer MH. * *p* < 0.05, ** *p* < 0.01, *** *p* < 0.001.

## Data Availability

The data presented in this study are available on request from the corresponding author by completion of the research project. The data are not publicly available due to ongoing publication of the project.

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
