# Peer review of "Video-Based Communication and Its Association with Loneliness, Mental Health and Quality of Life among Older People during the COVID-19 Outbreak"

_ijerph, 2021, doi:10.3390/ijerph18126284_

Round 1

Reviewer 1 Report

the authors addressed an important question during this health crisis, and I especially appreciate the authors for their efforts to collect cross-country nations. I encourage the authors go in depth in their review of the literature regarding interpersonal interactions via video-based media and mental health, which will make the paper stronger. The main concern I had was twofold. First, why did the authors choose to collect data from the four countries? Is the rationale simply that the research team consists of researchers from the four countries? It is important to have a more compelling rationale. Second, did the authors examine any country differences in the use of video-based media, loneliness, mental health, etc? Given that the four countries had different COVID-19 outbreak timeline, it is appropriate to separate the analyses or control for nationality at the minimum. 

Reviewer 2 Report

The present manuscript (MS) reports use of video-based communication technology (VBCT) amongst elderly people and associations with subjective loneliness, mental health, and quality of life. The data was collected in April/May 2020, which denotes the global pandemic outbreak of COVID-19, which widely entailed lockdown measures and social distancing.

The MS is well written and easy to read, raises strong arguments and ideas, and touches upon an important, perhaps easily overlooked topic within society: the mental well-being of elderly people who are impacted by restrictions that impair social contact, which can be met by engaging with communication technology.

I would like to raise some comments about methodology and analyses, which hopefully will help to improve the manuscript. As it is now, I remain somewhat doubtful as to whether the results can point to clear, valuable directions, with much criticism already being addressed by the authors themselves. Perhaps my comments can be addressed by providing more information, data, or rerunning some analyses. If this is not possible because said data are not available, or new results do not provide further implications, then I am less enthusiastic about the potential of this MS.

How meaningful are the results?

The main findings are as follows: Use of VBCT (yes or no) was more common in 60-69 year olds than the 70+ group. For the 60-69 year olds, this use was associated with less loneliness and higher quality of life. In contrast, for participants aged 70+, use of VBCT was not associated with any psychological markers.

The authors conclude that age influences how VBCT is associated with loneliness, (mental health,) and quality of life. However, they also mention that what drives this effect may be that more people aged 60-69 were employed. This issue concerns my major criticism, which vastly impacts the meaningfulness of the results.

The fact that there were no significant differences between 60-69 and 70+ year olds concerning loneliness and quality of life, or even that slightly more mental health problems were present in the younger group, calls into question if the results are actually as interesting as suggested. Intuitively, I would want to suspect that VBCT may help alleviate loneliness, problems, and impairments to one’s quality of life, by providing social connections in these challenging times. However, this very interpretation becomes questionable because the effect between age groups may hinge on employment status: 60-69 year olds could just be more often required to use VCBT because of their job (p. 6, ll. 234-237), and thus are more socially integrated, (less mentally well maybe because of stress and changes to their routine), and if they manage to effectively use VCBT, they experience less loneliness and more quality of life. So it is perhaps not an effect of VCBT, but employment.

As the authors point out themselves, it is not clear why and with whom elderly people use and engage with VBCT – this information would have been very valuable. Also, the MS emphasizes the COVID-19 outbreak. But in fact, we don’t know if use of VBCT changed because of said outbreak, and if it has anything to do with the presented variables and associations. The role of social distancing (as mentioned in keywords and introduction) is not elucidated, nor were technology competence and familiarity controlled. The reader does not know if, and to what degree, elderly participants feel affected by the pandemic, restrictions, or social distancing, and in turn, if VCBT is a resource in this context.

Categorical (yes/no) assessment of VBCT use

Using a dichotomous variable (yes/no) to assess VBCT use leaves much in the dark – how often, how long, with whom do people communicate? Which period of time was addressed (in general, in the past few weeks, or since a pandemic has broken out)? In terms of interpretation, it does not offer information about “how often” people use VBCT (cf. p. 6, ll. 220-222), only if they do or do not: More people in the 60-69 year old group indicated that they use VBCT compared to those 70+ years old, but we don’t know if they use it often. (This may sound fussy, but it is a wrong conclusion in an important context, and very easily misunderstood by readers).

Regression

Typically, in regression analyses, model significance should be reported (meaning, e.g., please provide information if the 6.6% of loneliness variance explained by the factors in 60-69 year olds is a significant portion of variance). By what method was regression performed (enter, stepwise, …) and how did it handle covariates? (E.g., by including the set of demographic variables in a first step, then entering VBCT in a second step.) Did you test if your regression results differ between those who are employed vs. those who are not (instead of, or in addition to, age groups)? I think it would also be useful to learn about correlations between loneliness, mental health, and quality of life. Also, please evaluate that your participants seem to fare well – they are not particularly lonely, are not impaired by mental health problems, and enjoy a good quality of life.

Round 2

Reviewer 2 Report

I appreciate the responses and enjoyed reading the revised manuscript. I think that the authors have made constructive amendments that provide some helpful information.

I find my points are sufficiently addressed. Especially the additions concerning pre-pandemic levels of loneliness, QOL and mental health, as well as the explanation regarding employment status are beneficial. I have just one small recommendation in response to the employment issue:

R2: The main findings are as follows: Use of VBCT (yes or no) was more common in 60-69 year olds than the 70+ group. For the 60-69 year olds, this use was associated with less loneliness and higher quality of life. In contrast, for participants aged 70+, use of VBCT was not associated with any psychological markers. The authors conclude that age influences how VBCT is associated with loneliness, (mental health,) and quality of life. However, they also mention that what drives this effect may be that more people aged 60-69 were employed. This issue concerns my major criticism, which vastly impacts the meaningfulness of the results.

Authors: Among those aged 60-69, employment was related to better mental health. Employment was not significantly associated with loneliness and quality of life. In this age group, however, even when controlling for employment, use of VBCT was associated with less loneliness and better quality of life. Thus, the associations between VBCT and loneliness/quality of life were still present after adjustment and were stronger than the associations between employment and these outcomes (see Table 2). Thus, while employment might drive a difference in the use of VBCT between age groups (as stated in the Discussion section), it did not drive the difference in associations between VBCT and outcomes. We have modified the Conclusion section to reflect the above stated views.

R2 response: Thank you for looking into this! I think you should consider emphasising that “even when controlling for employment, use of VBCT was associated with less loneliness and better quality of life / [employment status] did not drive the difference in associations between VBCT and outcomes” more explicitly early in the discussion section.